# Protocadherin Gamma C3 (PCDHGC3) Is Strongly Expressed in Glioblastoma and Its High Expression Is Associated with Longer Progression-Free Survival of Patients

**DOI:** 10.3390/ijms23158101

**Published:** 2022-07-22

**Authors:** Jonas Feldheim, David Wend, Mara J. Lauer, Camelia M. Monoranu, Martin Glas, Christoph Kleinschnitz, Ralf-Ingo Ernestus, Barbara M. Braunger, Patrick Meybohm, Carsten Hagemann, Malgorzata Burek

**Affiliations:** 1Section Experimental Neurosurgery, Department of Neurosurgery, University Hospital Würzburg, 97080 Würzburg, Germany; jonas.feldheim@uk-essen.de (J.F.); david.wend@stud-mail.uni-wuerzburg.de (D.W.); ernestus_r@ukw.de (R.-I.E.); 2Center for Translational Neuro- and Behavioral Sciences, University Hospital Essen, 45147 Essen, Germany; martin.glas@uk-essen.de (M.G.); christoph.kleinschnitz@uk-essen.de (C.K.); 3Division of Clinical Neurooncology, Department of Neurology, University Hospital Essen, 45147 Essen, Germany; 4Department of Anaesthesiology, Intensive Care, Emergency and Pain Medicine, University Hospital Würzburg, 97080 Würzburg, Germany; lauer_m2@ukw.de (M.J.L.); meybohm_p@ukw.de (P.M.); 5Graduate School of Life Sciences, Julius-Maximilians-University Würzburg, 97074 Würzburg, Germany; 6Department of Neuropathology, Institute of Pathology, Julius-Maximilians-University Würzburg, 97080 Würzburg, Germany; camelia-maria.monoranu@uni-wuerzburg.de; 7Institute of Anatomy and Cell Biology, Julius-Maximilians-University Würzburg, 97070 Würzburg, Germany; barbara.braunger@uni-wuerzburg.de

**Keywords:** glioblastoma multiforme, glioma, astrocytoma, recurrence, relapse, mRNA, protein, brain, expression, PCDHGC3, WNT signaling

## Abstract

Protocadherins (PCDHs) belong to the cadherin superfamily and represent the largest subgroup of calcium-dependent adhesion molecules. In the genome, most PCDHs are arranged in three clusters, *α*, *β*, and *γ* on chromosome 5q31. PCDHs are highly expressed in the central nervous system (CNS). Several PCDHs have tumor suppressor functions, but their individual role in primary brain tumors has not yet been elucidated. Here, we examined the mRNA expression of PCDHGC3, a member of the *PCDHγ* cluster, in non-cancerous brain tissue and in gliomas of different World Health Organization (WHO) grades and correlated it with the clinical data of the patients. We generated a PCDHGC3 knockout U343 cell line and examined its growth rate and migration in a wound healing assay. We showed that PCDHGC3 mRNA and protein were significantly overexpressed in glioma tissue compared to a non-cancerous brain specimen. This could be confirmed in glioma cell lines. High PCDHGC3 mRNA expression correlated with longer progression-free survival (PFS) in glioma patients. PCDHGC3 knockout in U343 resulted in a slower growth rate but a significantly faster migration rate in the wound healing assay and decreased the expression of several genes involved in WNT signaling. PCDHGC3 expression should therefore be further investigated as a PFS-marker in gliomas. However, more studies are needed to elucidate the molecular mechanisms underlying the PCDHGC3 effects.

## 1. Introduction

The WHO classification divides gliomas, brain tumors of the central nervous system (CNS), into four different grades, which can be defined based on their histological differentiation, anaplasia, and aggressiveness. Molecular genetic factors were first introduced into the WHO classification in 2016, and their impact to distinguish between tumor types and subtypes has increased in recent years [1,2]. An important molecular biological marker is isocitrate dehydrogenase (IDH), which is present either in its wild type (IDHwt) or in a mutated form (IDHmut). Mutations occur predominantly in the less malignant astrocytomas and oligodendrogliomas, WHO grade 2 and 3 (gliomas grade 2/3), and these patients often have a better prognosis than patients with IDHwt [3,4,5]. In the recent WHO classification of 2021, the differences between IDHwt and IDHmut gliomas were addressed by weighing the impact of an IDH mutation higher than the morphological criteria that built the backbone of brain tumor classifications for almost a century. Astrocytic tumors with necrosis or microvascular proliferation and IDH mutation are therefore no longer classified as glioblastomas (GBM, WHO grade 4), but as astrocytoma IDHmut, WHO grade 4 [6]. In contrast, a large proportion of the former grade 2 or 3 astrocytomas without an IDH mutation have been reported to be molecularly closely related to GBM and might be treated accordingly [7,8].

Despite standard therapy, which consists of tumor resection with subsequent radiation in combination with temozolomide chemotherapy followed by adjuvant temozolomide treatment, the median survival of GBM patients is only 15–20 months [9,10]. Patients usually develop a relapse within the first year after resection, which again is resected, irradiated, and treated with chemotherapy. Therefore, more efficient treatment options are urgently needed. The advancements in genomics and proteomics have allowed researchers to gather prominent molecular biomarkers. Understanding of the role and mechanism of these biomarkers in GBM tumorigenesis is crucial for improved treatment options. Thus, the characterization of new disease markers may contribute to the development of successful therapies and higher survival rates for GBM patients [11].

Cadherins are a family of proteins that play an important role in cell–cell contacts and in calcium-dependent cell adhesion. Over 100 different members of the cadherin superfamily are now known [12,13]. With over 80 different members, protocadherins (PCDHs) represent the largest subgroup of this superfamily [14]. PCDHs are mainly expressed in the nervous system [15,16]. They have six or seven extracellular domains that differ greatly in their sequence from those of the classic cadherins [17]. Due to their genetic organization, PCDHs can be divided into two different subgroups, the non-clustered PCDHs and the clustered PCDHs, which, due to their specific gene structure, can in turn be subdivided into the gene clusters *PCDHα*, *PCDHβ*, and *PCDHγ*, all located on chromosome 5q31 [17,18,19].

PCDHGC3, a member of the PCDHγ cluster, has been described in previous studies as a tumor suppressor that promotes the apoptosis of tumor cells, as well as suppresses the Wnt- and mTOR-signaling pathways, which negatively affects the growth of various tumors such as Wilms tumors, breast cancer, or prostate cancer [20]. When 53 different members of the PCDH family were examined in colorectal adenomas and carcinomas, mostly asynchronous hypermethylations were found [20]. For *PCDHGC3*, hypermethylation was detected in 17 of 28 carcinomas (60.7%). PCDHGC3 has the highest expression in non-cancerous colonic epithelium and there is a connection between the methylation of PCDHGC3 and its expression in tumors. Overexpression of PCDHGC3 in colon cancer cell lines resulted in a decreased cell proliferation and the ability to form colonies. In a Wilms tumor, a malignant tumor of the kidney, and in prostate cancer, the *PCDHGC3* promoter was found unmethylated, while in breast cancer, hypermethylation of the *PCDHGC3* promoter occurs, which leads to suppressed expression [21,22,23,24].

Therefore, the aim of this study was to examine PCDHGC3 expression in patient samples of gliomas with different WHO grades and to compare them with clinical parameters to determine whether the PCDHGC3 expression may serve as a prognostic biomarker. Previous RNA expression data of PCDHGC3, generated from the database of the “The Cancer Genome ATLAS” (TCGA) project, indicated increased expression in gliomas and melanomas. PCDHGC3 mRNA expression was correlated with the overall survival (OS) and progression free survival (PFS) of the patients. Our results can help to establish PCDHGC3 as a prognostic marker or therapeutic target in the future.

## 2. Results

### 2.1. The Cancer Genome ATLAS (TCGA) Data Show Increased PCDHGC3 mRNA Expression in Gliomas and Melanomas

The mRNA expression data of TCGA revealed increased PCDHGC3 mRNA expression in gliomas (regardless of the WHO grade) and melanomas. The other cancers analyzed, such as thyroid, lung, colorectal, head and neck, stomach, liver, and ovarian cancer, displayed very low PCDHGC3 expression (Figure 1a). Analysis of the Ivy Gap Dataset showed PCDHGC3 mRNA expression in areas of GBM with signs of microvascular proliferation (compared to the perinecrotic zone, infiltrating tumors, and cellular tumors; *p* < 0.05) or with hyperplastic blood vessels (compared to infiltrating tumor cells and cellular tumors; *p* < 0.05) to be significantly decreased, whereas PCDHGC3 mRNA expression in the cellular part of GBMs appeared to be significantly enhanced (compared to leading edge, pseudopalisading cells, hyperplastic blood vessels, and microvascular proliferation; *p* < 0.05) (Figure 1b).

### 2.2. Patient Cohorts Characteristics

The clinical data as well as the course of the disease and therapy could be collected for 46 of the 60 GBM patients of our own collective that were tested for their PCDHGC3 mRNA expression (Table 1). All patients were treated at the Department for Neurosurgery, University Hospital Würzburg between January 2011 and December 2013. For some patients, there were no data on O^6^-Methylguanin-DNA-Methyltransferase (MGMT) promoter methylation (*n* = 24), surgery (*n* = 15), radiation therapy and chemotherapy (*n* = 9), or on relapse (*n* = 7) available (Table 1). In the clinical course of 6 patients, we found significant confounders on survival and therefore decided to exclude them from the survival analyses. Table 2 shows the patient characteristics determined for gliomas grade 2/3 (*n* = 20). Our cohort of gliomas grade 2/3 consisted exclusively of tumors that were classified as astrocytoma IDHmut, CNS WHO grade 2 at the time of their initial diagnosis. Samples that were classified in another group according to the recent WHO classification were excluded [2]. However, due to limitations of sample quantity, we could not perform re-classification, including molecular diagnostics, on all samples and therefore chose the less specific terminology “gliomas grade 2/3”.

### 2.3. PCDHGC3 mRNA Expression in Gliomas Grade 2/3 and GBM-Subtypes

PCDHGC3 mRNA overexpression was found in gliomas grade 2/3 (median 8.41-fold, *p* < 0.001) and GBM (median 4.29-fold, *p* = 0.005) compared to non-cancerous brain samples. No significant difference was found between the different brain tumor entities (Figure 2a). The subgroup analysis of GBM patients covered 17 patients with local GBM and subsequent local recurrence, 10 patients with local GBM and subsequent multifocal recurrence, and 16 patients with primary multifocal GBM. The statistical analysis of the PCDHGC3 mRNA expression in these GBM with different growth patterns compared to non-cancerous brain samples revealed significant differences for local GBM with later local recurrence (*p* = 0.029), as well as primary multifocal GBM (*p* = 0.043) (Figure 2b). There were no statistical differences for patients with local GBM with a multifocal growth pattern in recurrence (*p* = 0.589). The comparison of the GBM subgroups with each other did not reveal any significant differences (*p* > 0.05).

### 2.4. Kaplan-Meier Analyses

Survival curves were generated for 40 GBM patients in relation to their OS (Figure 3a) and for 35 GBM patients in relation to their PFS (Figure 3b). We decided to exclude six of the original 46 patients from survival analyses, as we identified significant external confounders (e.g., severe complications unrelated to GBM). An additional five patients did not match the radiological criteria for tumor progress before their decease and were therefore excluded from the PFS analysis. The OS was also examined for 20 patients with gliomas grade 2/3 (Figure 3c). With regard to the OS of GBM patients, no correlation could be found for those with high or low PCDHGC3 expression (*p* = 0.790). However, the results for the PFS of GBM patients in comparison to PCDHGC3 expression were statistically significant. Patients with low PCDHGC3 expression had a significantly shorter median PFS of 7 months compared to patients with high PCDHGC3 expression, who had a median PFS of 12 months (*p* = 0.016) (Figure 3b). In a multivariable cox proportional hazards model, age, extent of resection, MGMT promoter methylation, and interestingly also the PCDHGC3 mRNA expression proved to be independent prognostic factors for the OS of GBM patients (Table 3), while none of the included characteristics were predictive for GBM patients’ PFS (*p* > 0.05). The OS of patients with gliomas grade 2/3 was not significantly different (37 vs. 72 months, *p* = 0.168). Despite the lack of significance, a clear advantage for patients with high PCDHGC3 expression was identified over a period of 72 months, while the median OS for patients with low expression was 37 months. After 72 months, the curves crossed, which means that all patients with high PCDHGC3 expression died (Figure 3c).

### 2.5. Correlation of Clinical Data with PCDHGC3 Expression

To determine possible subgroups with especially high or low PCDHGC3 mRNA expression, we performed further tests for correlations or uneven distribution. Apart from the PFS shown in Figure 3b (*p* = 0.018), no further correlations could be determined for GBM patients. The PCDHGC3 mRNA expression did not correlate with sex, MGMT promoter methylation, overall survival, tumor growth pattern, localization of tumor, tumor volume, Ki-67-staining, or type of therapy in GBM. However, for glioma grade 2/3 patients, a significant association between PCDHGC3 mRNA expression level and OS was found when the Wilcoxon-Mann-Whitney test was used (*p* = 0.022) (Table 4).

### 2.6. PCDHGC3 Protein Level Is Increased in GBM

Protein lysates were isolated from randomly selected samples of non-cancerous controls and GBM, followed by a Western blot analysis using a specific anti-PCDHGC3 antibody (Figure 4a). A statistically significant increase in PCDHGC3 protein levels was detected in GBM samples compared to non-cancerous brain samples (Figure 4b).

### 2.7. Deletion of PCDHGC3 in a GBM Cell Line Results in Faster Migration of Knockout Cells

Our results indicated that high PCDHGC3 mRNA expression levels were associated with longer PSF and might be an independent predictor for OS, as suggested by the multivariable cox proportional hazard model. We therefore tested widely used glioma cell lines GaMG, U87, U138, and U343 for PCDHGC3 protein expression (Figure 5a). Densitometric quantification revealed high levels of PCDHGC3 protein expression in U343, which thus was selected to generate a PCDHC3 knockout using the CRISPR/Cas9 method. Wild type (WT) U343 cells expressed high levels of PCDHGC3 (Figure 5a,b), while no PCDHC3 protein could be detected in the knockout (KO) cell line (Figure 5b). Next, we compared the metabolic activity corresponding to the cell number of WT and KO cells by MTT assay. The KO cells showed significantly lower cell numbers compared to WT cells (Figure 5c), but they migrated faster in the wound healing assay (Figure 5d), indicating that the deletion of PCDHGC3 in this cell line leads to a more migratory and invasive phenotype. Further experiments are needed to elucidate the proliferation and invasiveness of KO cells.

### 2.8. Genes Involved in WNT (Wingless/Integrated) Signaling Are Partially Downregulated in PCDHGC3 KO U343 Cells

WNT signaling regulates important cellular processes and is often overactive in GBM, contributing to GBM proliferation and invasiveness. We therefore measured the mRNA expression levels of several genes involved in WNT signaling in WT and PCDHGC3 KO U343 cells (Figure 6). PCDHGC3 KO showed significantly lower mRNA expressions of Fizzled Class Receptor 9 and 10 (FZD9, 10), LDL Receptor Related Protein 6 (LRP6), and WNT Family Member 6 (WNT6) compared to WT. β-Catenin (CTNNB1), Dickkopf WNT Signaling Pathway Inhibitor (DKK1), FZD2, and LRP5 showed no changes in mRNA expression.

## 3. Discussion

The expression data from this work show for the first time that gliomas exhibited increased PCDHGC3 mRNA expression compared to non-cancerous brain samples. This result corresponds to the data from the TCGA database, which indicate high expression in gliomas. Interestingly, when different GBM areas were analyzed, PCDHGC3 mRNA expression was significantly reduced in areas of hyperplastic blood vessels in cellular tumors and in areas of microvascular proliferation. This is consistent with our recently published results with brain microvascular endothelial cells, in which PCDHGC3 knockout resulted in increased endothelial proliferation [26,27]. However, no significant differences of mRNA expression could be found between different glioma types. Although GBM displayed a higher expression compared to gliomas grade 2/3, this turned out to be a statistically insignificant tendency. Significant expression differences were found for the GBM subgroups of local GBM with later local recurrence, as well as for primary multifocal GBM compared to non-cancerous brain samples. The only exception was local GBM with later multifocal recurrence, most probably due to the small sample size (*n* = 10) and high fluctuations in mRNA expression. If one looks at the medians of the GBM and gliomas grade 2/3 subgroups, it can be seen that these are very close to each other, pointing to a similar PCDHGC3 expression. These similar medians also indicate that the expression of PCDHGC3 did not correlate to the size or tendency to the multifocal growth of the gliomas examined, nor to any of the other clinical characteristics. Therefore, we were not able to determine glioma subgroups of high or low PCDHGC3 mRNA expression and concluded that PCDHGC3 serves no value as a predictive biomarker for these characteristics.

Overall, PCDHGC3 mRNA was significantly lower expressed in the non-cancerous brain compared to the tumors, which distinguishes it from other members of the Protocadherin superfamily such as PCDH8 or PCDH9. These were described to have tumor suppressor properties and to display greatly reduced expression in gliomas compared to non-cancerous brain samples [28,29]. Despite the small sample size of the non-cancerous brain samples (*n* = 10), these data can be linked very well to other research work in which similar cohorts (*n* = 6 for non-cancerous brain) were available [30]. A recent publication showed that PCDHGC3 is the only isoform within its cluster that inhibits the oncogenic Wnt signaling pathway [31]. Therefore, further experiments should clarify whether the Wnt signaling pathway is downregulated in PCDHGC3 overexpressing gliomas and whether the silencing of PCDHGC3 would lead to increased tumor growth, as has been shown for PCDH8 [29].

To the best of our knowledge, these are the first data on the impact of PCDHGC3 on the clinical course of patients. A TCGA database-based Kaplan–Meier analysis comparing “low” and “high” PCDHGC3 mRNA expressing gliomas, as separated by the median expression, revealed no significant correlation of PCDHGC3 expression and patients’ survival (*p* = 0.43). However, this Kaplan–Meier plot showed the survival time for all glioma patients combined, regardless of their WHO grading. While pilocytic or diffuse astrocytomas WHO grade 1 or grade 2 are easily treatable and a survival time of several years to decades is not uncommon, GBM patients have a median survival time of only 15–20 months [9,10]. Thus, carrying out a combined survival time analysis of all glioma patients is not conclusive. It is much more appropriate to consider the WHO grading for such analyses, as has been performed in this work.

The PFS of GBM patients with high PCDHGC3 mRNA expression was significantly higher than that of patients with low PCDHGC3 expression (12 vs. 7 months, *p* = 0.016). However, it is necessary to take a closer look at the collection of the PFS data to evaluate the impact of this statement. Data collection began after the tumor resection. Patients were treated with radiation therapy combined with temozolomide chemotherapy and adjuvant temozolomide chemotherapy [32]. The majority of the patients were then discharged from the hospital depending on their state of health. The first MRI follow-up examinations took place 21–28 days later, with repetitions at 3-month intervals [32]. In case tumor progression appeared shortly after such a 3-month screening and was discovered at the next screening another 3 months later, it would shift the PFS-period considerably. However, patients were encouraged to arrange for follow-up examinations early if they observe new or worsened clinical symptoms. Nevertheless, even if such bias and the comparatively low cohort size (*n* = 20) are considered, a 5 months improved PFS could mean an enormous increase in quality of life for the patient. Interestingly, in a multivariable model with known prognostic factors, PCDHGC3 proved to offer an independent predictive value for OS. In contrast, none of the included factors, not even the ones known to yield prognostic value, significantly predicted GBM patients’, which is most likely due to the comparably small sample number.

While we consider our observations on PCDHGC3 association with patients’ OS and PFS interesting and noteworthy, they should be interpreted with caution at this stage, and we encourage further studies to verify them and determine if PCDHGC3 should be considered as a prognostic marker or as a therapeutic target in the future.

The OS of GBM and gliomas grade 2/3 patients was not dependent on PCDHGC3 mRNA expression. These data suggested that PCDHGC3 does not have tumor suppressor properties, at least in gliomas, in contrast to other PCDHs such as PCDH9 [28,30]. Nevertheless, the Wilcoxon Mann–Whitney test revealed an association of PCDHGC3 and the OS of patients with gliomas grade 2/3 (*p* = 0.022). This discrepancy to the non-significant Kaplan–Meier analysis might be due to the crossing of both survival curves, which was caused by 2 of the 10 patients (20%) in the high PCDHGC3 expression group. However, 60% of the latter patients had an OS almost twice as long (72 months) as the patients in the low-expression group (37 months). A weakness of this analysis is the low cohort size, which should be enlarged in future analyses, also with regard to a better comparability to the GBM patients’ survival time. These results were consistent with our analysis of PCDHGC3 knockout glioma cells, which showed a slower growth rate but a much faster migration rate. Altered expression of genes involved in WNT signaling in PCDHGC3 KO U343 cells may be involved in this phenotype.

For the first time, we analyzed PCDHGC3 expression in gliomas under consideration of their WHO grading. The overexpression of PCDHGC3 in gliomas grade 2/3 and GBM compared to non-cancerous brain samples was analyzed on both the mRNA and protein level. The correlation with clinical parameters indicated that PCDHGC3 might serve as a useful future marker for PFS in GBM patients. PCDHGC3 KO leads to changes in phenotype and gene expression in the U343 cell line, indicating its distinct role in GBM pathology.

## 4. Materials and Methods

### 4.1. Patient Cohort

The patient cohort of gliomas was compiled in a previous study [33]. The patients included in this study stated their written informed consent in accordance with the International Conference on Harmonization, the declaration of Helsinki as approved by the Institutional Review Board of the University of Würzburg (#103/14). In addition, we obtained autopsy/biopsy samples of non-pathological brain tissue from the Brain Bank of the Department of Neuropathology, Institute of Pathology, University of Würzburg, Germany (approval #78/99). Tissue samples from control autopsies and biopsies without proven central nervous system (CNS) pathologies (corresponding to non-cancerous brain) served as reference samples (*n* = 10). In order to investigate correlations, clinical data were retrospectively collected for the gliomas grade 2/3, as well as for the GBM patients. Due to the external treatment of some patients, insufficient sample material, or missing data sets, the complete clinical data set could not be collected for every patient. Therefore, not all patients could be included in every analysis.

### 4.2. Quantitative Real-Time PCR

To detect the expression level of PCDHGC3, qPCR was carried out. All samples used were already available as isolated RNA (−80 °C) from a previous study but still had to be transcribed into cDNA [33]. The RNA was gently thawed on ice, and its concentration was measured using NanoDrop (Thermo Fisher Scientific, Waltham, MA, USA). The transcription of mRNA into cDNA was carried out using the High-capacity cDNA reverse transcription kit (Thermo Fisher Scientific, Waltham, MA, USA). RNA samples were used in a concentration of 5 ng/μL for cDNA synthesis. The cDNA was stored at −20 °C until further use. The TaqMan^®^ Fast Advanced Master Mix (Thermo Fisher Scientific, Waltham, MA, USA) was used for qPCR. cDNA samples were gently thawed on ice. All qPCR runs were pipetted into 96-well plates. The total reaction volume of 20 μL consisted of 10 μL Master Mix, 5 μL ultra-pure water, 4 μL cDNA, 1 μL of the endogenous control (18S RNA), and 1 μL of the PCDHGC3 TaqMan^®^ gene expression assay (Thermo Fisher Scientific, Waltham, MA, USA). Interactions between the probe and endogenous control were excluded by preliminary tests. Each patient sample was analyzed in triplets. The 96-well plate was sealed with self-adhesive film and then centrifuged at 200× *g* for a few seconds. The plate was transferred to the StepOnePlus™ Real-Time PCR System (Applied Biosystems, Waltham, MA, USA) and ran with the parameters suggested by the supplier. All qPCR runs were checked for validity. This was only given for samples whose 18S RNA control threshold cycle values were within 8.0 and 18.0. Samples outside this range were excluded via the StepOnePlus™ software v2.3 (Applied Biosystems, Waltham, MA, USA). The qPCR runs were then imported into ExpressionSuite Software (Thermo Fisher Scientific, Waltham, MA, USA) in order to combine them into a single table. Finally, the clinical patient data were assigned to the respective sample designations.

### 4.3. Western Blot

Proteins were isolated from the same brain samples as the RNA. After the centrifugation of samples with TRIzol^®^ Reagent (Thermo Fisher Scientific, Waltham, MA, USA) and after removing the aqueous phase for RNA isolation, the interface and organic phenol-chloroform phase were used for protein precipitation (normal brain biopsies, gliomas grade 2/3, and GBM samples) according to the manufacturer’s instructions. The total protein concentration was determined using a Pierce^TM^ 660 nm Protein Assay Reagent (Thermo Fisher Scientific, Waltham, MA, USA) according to the manufacturer’s instructions. An amount of 40 μg of total protein lysate was loaded on 10% SDS-PAGE gels. Protein-specific signals were detected using a rabbit anti-PcdhgC3 antibody (1:10,000, generously provided by M. Frank, University of Freiburg, Germany), which binds to both mouse and human PCDHGC3 [26,34]. A mouse monoclonal anti-GAPDH antibody (1:1000, Millipore, Burlington, MA, USA) was used as the endogenous control. The bands were quantified by densitometric analysis (Image J 1.53c Software, NIH). The PCDHGC3 protein expression level in widely used glioma cell lines GaMG, U87, U138 and U343 was determined by Western blot, using the rabbit anti-PcdhgC3 antibody as described above [26,34]. A mouse monoclonal anti-b-actin antibody (Sigma Aldrich, St. Louis, MO, USA) was used as an endogenous control.

### 4.4. Generation of the PCDHGC3 Knockout Cell Line

U343 cells expressing high levels of PCDHGC3 were selected for the construction of PCDHGC3 knockout. U343 cells were co-transfected with PCDHGC3 CRISPR/Cas9 and HDR vectors (Santa Cruz Biotechnology, Dallas, TX, USA), as described previously [26,27]. Transfected clones were selected with 3 µg/mL puromycin, and knockout was verified by Western blot as described above. RNA was isolated from WT and PCDHGC3 KO U343 cells as previously described [35,36] using the RNA isolation kit NucleoSpin^®^ RNA (Machery-Nagel, Düren, Germany) according to manufacturer’s instructions. Total RNA (500 ng) was reverse transcribed using the High-Capacity cDNA Reverse Transcription Kit (Thermo Fisher Scientific, Waltham, MA, USA). The TaqMan probes Hs00355045_m1 (CTNNB1), Hs00183740_m1 (DKK1), Hs00361432_s1 (FZD2), Hs00268954_s1 (FZD9), Hs04999826_s1 (FZD10), Hs01124561_m1 (LRP5), Hs00233945_m1 (LRP6), and Hs00362452_m1 (WNT6) were used with the TaqMan^®^ Fast Advanced Master Mix in the QuanStudio 7 flex Fast Real-Time PCR System (Thermo Fisher Scientific, Waltham, MA, USA). Calnexin (CANX, Hs01558409_m1) and β-actin (ACTB, Hs00357333_g1) were used as endogenous controls. Relative expression was calculated by the comparative Ct method using QuantStudio^TM^ Real-Time PCR Software v1.7.2 (Thermo Fisher Scientific, Waltham, MA, USA).

### 4.5. Wound Healing Assay

The wound healing assay was performed as previously described [37,38]. Briefly, wild type and PCDHGC3 knockout U343 cells were grown in μ-Dishes (Ibidi, Gräfelfing, Germany) in silicone inserts separated by a 500-µm cell-free area for 24 h. The silicone inserts were removed, and images corresponding to time 0 h were taken. Cells were allowed to migrate into the cell-free area for 48 h followed by microscopic documentation with a Keyence BZ9000 microscope (Keyence, Osaka, Japan). The percentage of wound closure was calculated by measuring the cell-free area at times 0 h and 48 h with BZ-II-Analyzer software (Keyence, Osaka, Japan).

### 4.6. Cell Viability Measurement

Cell viability and cellular metabolic activity was determined by the 3-(4,5-dimethylthiazolyl-2)-2,5-diphenyltetrazolium bromide (MTT) assay as previously described [36]. Wild type and PCDHGC3 knockout U343 cells were seeded in 96-well flat-bottom plates at 3 × 10^5^ cells/well. After 24 h, 100 μL of MTT solution (1 mg/mL, Sigma Aldrich, St. Louis, MO, USA) in DMEM without phenol red was added to each well and incubated at 37 °C for 4 h. After incubation, formazan crystals were solubilized with 100 µL of isopropanol, followed by an absorbance measurement at 560 nm and 690 nm using a Tecan Microplate Reader (Tecan, Männedorf, Switzerland).

### 4.7. Statistical Analysis

Statistical analysis was performed using SPSS Statistics 23 (IBM, Armonk, NY, USA) or GraphPad Prism 9 (GraphPad Software, San Diego, CA, USA). The differences in the expression of PCDHGC3 mRNA were determined using the ΔΔ threshold cycle (CT) algorithm [39]. First, the ΔCT value was calculated (CT target gene—CT reference gene). The ΔCT values of the non-cancerous brain samples were then averaged. The ΔΔCT value was determined by subtracting the ΔCT values of the tumor samples from the ΔCT value of the mean non-cancerous brain values. Relative expression differences (RQ) between tumor samples and non-cancerous brain samples, which were normalized to a reference gene (18S RNA), resulted from the arithmetic formula 2^−ΔΔCT^. The data generated were then statistically correlated with various parameters that were obtained as described elsewhere [33,40].

The PCDHGC3 expression of gliomas grade 2/3 and GBM as well as subgroup analyses of gliomas grade 2/3 and GBM were compared in pairs by Kruskal–Wallis (post-hoc test: Dunn’s test with Bonferroni correction). To investigate correlations between the RQ (GBM and gliomas grade 2/3 3 patients) and the clinical data, the Spearman rank correlation was carried out. The Wilcoxon Mann–Whitney test was used for binomial data (gender, MGMT status, relapse, radiation therapy, and chemotherapy). Kaplan–Meier analyses for gliomas grade 2/3 and GBM patients were carried out exclusively with patients whose clinical course could be fully reconstructed. The mRNA expression data available from these patients were divided into two categories based on the median. Patients whose mRNA expression was below the median were assigned to the category “low PCDHGC3 expression”, and patients with mRNA expression above the median were assigned to the category “high PCDHGC3 expression”. The Kaplan–Meier analysis was then determined via Cox regression. A multivariable Cox proportional hazard model (backward selection) was calculated for the OS and PFS of GBM patients. We did not perform a preliminary univariable analysis but chose to include the PCDHGC3 mRNA expression with co-variates known to be prognostic factors (age, MGMT promoter methylation, ECOG [0 vs. ≥1], extent of resection (biopsy and incomplete resection vs. complete resection), and tumor volume). We refrained from presenting a multivariable model of gliomas grade 2/3 due to the low number of samples.

Apart from our own patient collective, we analyzed PCDHGC3 mRNA expression in different areas of GBM with the Ivy Gap Database [25] (https://glioblastoma.allninstitute.org (accessed on 31 March 2022)) as describe elsewhere [40].

## Figures and Tables

**Figure 1 ijms-23-08101-f001:**
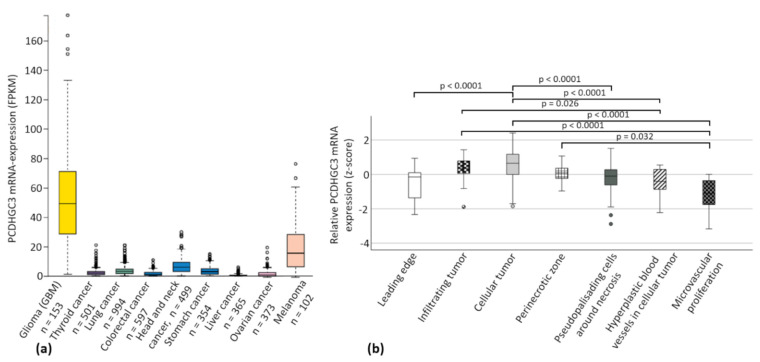
PCDHGC3 mRNA expression in different tumor entities and regions of GBM. (**a**) Different tumor types of the TCGA database were screened for PCDHGC3 mRNA expression (modified from The Cancer Genome Atlas 2020, National Cancer Institute). FPKM = Fragments Per Kilobase of transcript per Million mapped reads. (**b**) PCDHGC3 mRNA expression was compared in different areas of GBM by analyzing the Ivy Gap Dataset [25]. Different areas included leading edge (*n* = 19), infiltrating tumor (*n* = 24), cellular tumor (*n* = 111), perinecrotic zone (*n* = 26), pseudopalisading cells around necrosis (*n* = 40), hyperplastic blood vessels in cellular tumor (*n* = 22), and microvascular proliferation areas (*n* = 14). Kruskal–Wallis test, post-hoc: Dunn’s test, correction of the significance according to Bonferroni. Circles mark statistical outliers.

**Figure 2 ijms-23-08101-f002:**
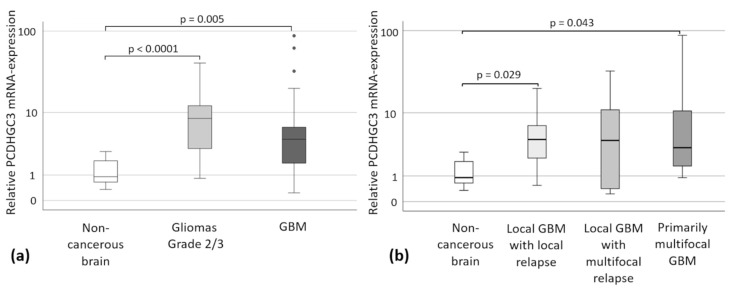
PCDHGC3 mRNA expression of gliomas grade 2/3 and GBM subgroups. Circles mark statistical outliers. Kruskal–Wallis test, post-hoc: Dunn’s test, correction of the significance according to Bonferroni. Medians and quartiles are shown. Error bars mark the confidence interval of 95%. (**a**) PCDHGC3 mRNA expression of gliomas grade 2/3 (*n* = 20) and GBM (*n* = 60, including the GBM without completely collected clinical data) compared to non-cancerous brain samples (*n* = 10). Circles mark statistical outliers. (**b**) PCDHGC3 mRNA expression of the GBM subgroups: local GBM with later local recurrence (*n* = 17), local GBM with later multifocal recurrence (*n* = 10), and primary multifocal GBM (*n* = 16) compared to non-cancerous brain samples (*n* = 10). Kruskal–Wallis test, post-hoc: Dunn’s test, correction of the significance according to Bonferroni.

**Figure 3 ijms-23-08101-f003:**
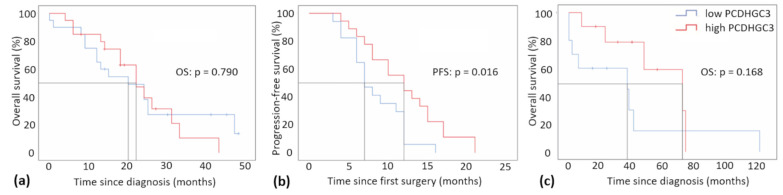
Kaplan–Meier plots of GBM patients and patients with gliomas grade 2/3 based on PCDHGC3 mRNA-expression. Samples were assigned either to the category “low” PCDHGC3 expression or “high” PCDHGC3 expression, based on the median of the mRNA expression. (**a**) Overall survival (OS) of GBM patients (*n* = 40, alignment with median = 22.0 months) in months since diagnosis (*p* = 0.790, median overall survival 20 vs. 22 months). (**b**) Progression-free survival of patients (PFS) with GBM (*n* = 35, alignment with the median = 10.0 months) since the first surgery in months (*p* = 0.016, median progression-free survival 7 vs. 12 months). (**c**) Overall survival (OS) of patients with gliomas grade 2/3 (*n* = 20, alignment with median = 41.0 months) in months since diagnosis (*p* = 0.168, median overall survival 37 vs. 72 months).

**Figure 4 ijms-23-08101-f004:**
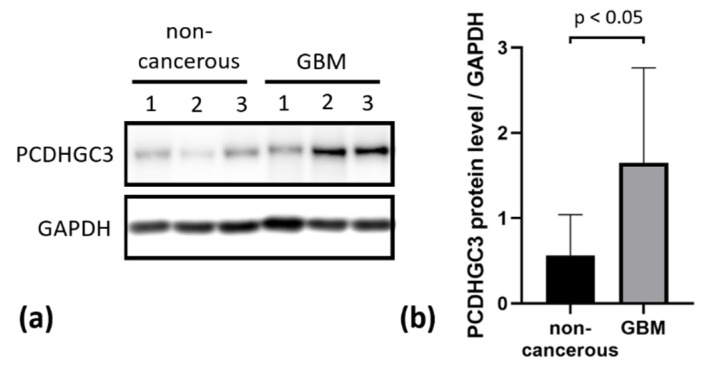
Western blot analysis of PCDHGC3 protein expression in non-cancerous and GBM brain samples. (**a**) Western blot and (**b**) densitometric quantification normalized to the endogenous control GAPDH, non-cancerous *n* = 7, GBM *n* = 12. Statistical significance was estimated using the Mann–Whitney test.

**Figure 5 ijms-23-08101-f005:**
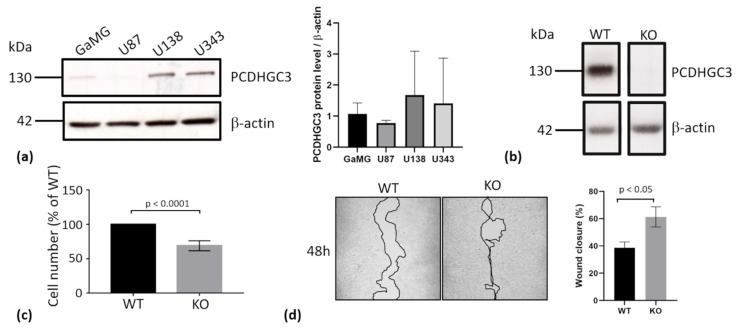
Deletion of PCDHGC3 in U343 cells results in a more invasive phenotype. (**a**) PCDHGC3 protein expression was determined by Western blot in glioma cell lines GaMG, U87, U138, and U343. The densitometric analysis of four passages of each cell line is shown on the right (*n* = 4). (**b**) U343 was chosen to generate a PCDHGC3 knockout. The wild type (WT) expressed high levels of the PCDHGC3 protein, while PCDHGC3 knockout (KO) U343 cells displayed a complete deletion of PCDHGC3, as shown by Western blot. (**c**) Cell viability and metabolic cellular activity corresponding to cell number were measured by MTT assay and expressed as % of WT cells. (**d**) PCDHGC3 knockout (KO) U343 cells migrated faster than wild type (WT) cells in the wound healing assay (magnification 100×, cell borders are marked with a black line). The graph shows the mean percentage of wound closure at 48 h compared to 0 h and the standard deviation of three independent experiments.

**Figure 6 ijms-23-08101-f006:**
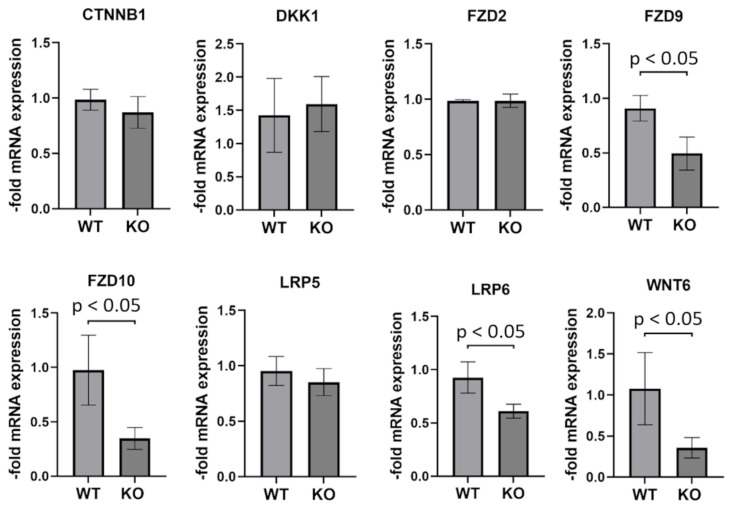
Genes involved in WNT signaling are partially downregulated in U343 PCDHGC3 knockout cells. Different passages of wild type (WT) and PCDHGC3 knockout (KO) U343 cells were used for RNA isolation and mRNA analysis by real-time PCR. Target gene expression was normalized to the endogenous control. Data are presented as mean ± standard deviation of *n* = 4. Statistical significance is indicated by *p* < 0.05, as measured by the Mann–Whitney test. CTNNB1: β-Catenin; DKK1: Dickkopf WNT Signaling Pathway Inhibitor; FZD2, 9, 10: Fizzled Class Receptor 2, 9, 10; LRP5, 6: LDL Receptor Related Protein 5, 6; WNT6: WNT Family Member 6.

**Table 1 ijms-23-08101-t001:** Clinical parameters of glioblastoma IDH wild type, CNS WHO grade 4 (GBM) patients (*n* = 46).

Patients’ Characteristics	
Sex	Female: 19/41.3% Male: 27/58.7%
Age (median, quartiles)	57.5 years (49.0–69.25 years)
ECOG (median, quartiles)	0 (0–1)
**Tumor Characteristics**	
Volume (median, quartile)	25.5 cm^3^ (15.3–54.2 cm^3^)
Tumor localization	Left hemisphere: 26/56.5% Right hemisphere: 17/37% Both hemispheres: 3/6.5%
Localization in the lobe of the brain	Frontal: 15/32.6% Occipital: 5/10.9% Temporal: 7/15.2% Parietal: 5/10.9% Multiple: 13/28.3% Cerebellar: 1/2.2%
MGMT promoter methylation	Unmethylated: 10/29.4% Methylated: 24/70.6%
Ki67 staining (median, quartile)	25% (20–30%)
**Therapy**	
Period between diagnosis and therapy (median, quartile)	6 days (4–13 days)
Surgical intervention	Biopsy: 6/13.3% Complete resection: 10/22.2% Incomplete resection: 29/64.4%
Radiotherapy	Yes: 43/93.0% No: 8/17.4%
Chemotherapy with temozolomide	Yes: 43/93.5% No: 8/17.7%
PFS (median, quartiles)	9 months (6–13 months)
Relapse	Local GBM (multifocal relapse): 13/28.3% Local GBM (local relapse): 26/56.5% Multifocal GBM: 7/15.2%
OS (median, quartiles)	18 months (12–25 months)

Given are the absolute numbers of the GBM patients in each group and the percentage of the analyzed population. ECOG = Eastern Cooperative Oncology Group score; MGMT = O^6^-methylguanine-DNA-methyltransferase; PFS = progression-free survival; OS = overall survival.

**Table 2 ijms-23-08101-t002:** Clinical parameters of gliomas grade 2/3 (*n* = 20).

Patient’s Characteristics	
Sex	Female: 7/35.0% Male: 13/65.0%
Age (median, quartile)	36.0 years (30.8–45.8 years)
OS (median, quartiles)	34.0 months (9.8–46.0 months)

Given are the absolute numbers of the specified gliomas grade 2/3 patients in each group and the percentage of the analyzed population. We classified the growth pattern according to the amount of brain lobes infiltrated by the tumor. OS = overall survival; gliomas grade 2/3 = isocitrate dehydrogenase (IDH)-mutated tumors with the histological appearance of WHO grade 2 and 3 gliomas.

**Table 3 ijms-23-08101-t003:** Multivariable Cox proportional hazard model for the OS of GBM patients.

	Hazard Ratio	95% Confidence Interval	*p*-Value
Age	1.115	1.046–1.189	0.001
MGMT promoter methylation	0.130	0.032–0.532	0.005
PCDHGC3 mRNA expression	1.038	1.008–1.068	0.012
Extent of resection	0.073	0.010–0.517	0.009

Multivariable cox proportional hazards model of the OS of GBM patients was determined via a stepwise backwards approach. PCDHGC3 mRNA expression was tested together with the known prognostic factors age, MGMT promoter methylation, extent of resection, ECOG, and tumor volume. ECOG and tumor volume were excluded in steps 1 and 2.

**Table 4 ijms-23-08101-t004:** Correlation and distribution of PCDHGC3 mRNA expression within glioma grade 2/3 patients and tumor characteristics.

Patient and Tumor Characteristics	Correlation Coefficient r	*p*-Value
Sex (female/male) *		0.703
Age	0.056	0.813
Tumor growth pattern *		>0.99
OS	0.509	0.022

Correlation and distribution of PCDHGC3 mRNA expression within selected patient and tumor characteristics were examined by non-parametric tests (Spearman’s Rho and Wilcoxon-Mann-Whitney test). If a correlation coefficient was given, variables were examined by Spearman’s Rho, otherwise Wilcoxon Mann–Whitney (*) test was performed. OS = overall survival.

## Data Availability

Data are contained within the article. Raw data can be obtained from the corresponding authors on reasonable request.

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
