# Peer review of "Protocadherin Gamma C3 (PCDHGC3) Is Strongly Expressed in Glioblastoma and Its High Expression Is Associated with Longer Progression-Free Survival of Patients"

_ijms, 2022, doi:10.3390/ijms23158101_

Round 1
Reviewer 1 Report
A brief summary:
The authors of this manuscript describe the prognostic role of the PCDHGC3 gene in glioblastoma (GBM). First, they performed a bioinformatics analysis by The Cancer Genome ATLAS (TCGA) data, showing that the expression of PCDHGC3 is elevated in glioma and melanoma. Survival curves showed that patients with low PCDHGC3 expression have short progression free survival. According to these results, the authors evaluated PCDHGC3 expression on different human GBM cell lines, demonstrating that the U343 cell line had the highest expression. This line was chosen to generate the PCDHC3 knockout (KO) and to evaluate cell proliferation and cell migration.
Broad comments:
Experimental evidence with PCDHGC3 KO cell line have serious flaws that cannot be included in the publication in the current version.
Specific comments:
11) In the abstract, it would be appropriate to specify the term PCDHGC3 as a member of one of the clusters.
2) Lines 81-82: authors declare “[..] to determine whether the PCDHGC3 expression may serve as a predictive biomarker”. A predictive biomarker gives information about the effect of a therapeutic intervention, but the authors did not consider any therapeutic association, therefore I suggest considering the aim of study at analyzing the PCDHGC3 as prognostic biomarker.
3) Line 189: the authors state “Our results indicate that high PCDHGC3 mRNA expression levels correlate with longer PSF and OS in patients”. Survival curves analysis did not demonstrate significance for OS. Therefore, I suggest changing this sentence.
4) The authors could perform a densitometric analysis of western blot to confirm the expression of the gene in GBM cell lines.
5) The authors show the MTT results plotting the number of cells in relation to the PCDHGC3 gene WT/KO. This result has serious errors because i) the absorbance data are not showed; ii) the MTT does not provide the number of cells, therefore the data are incorrect; iii) although this test is widespread for survival evaluation, it is not the ideal assay to evaluate the proliferative effect; indeed MTT is not reflecting necessarily cell proliferation and growth, but viable cell metabolism (Riss, T.L.; Moravec, R.A.; Niles, A.L.; Duellman, S.; Benink, H.A.; Worzella, T.J.; Minor, L. Cell Viability Assays. In Assay Guidance Manual, Markossian, S., Sittampalam, G.S., Grossman, A., Brimacombe, K., Arkin, M., Auld, D., Austin, C.P., Baell, J., Caaveiro, J.M.M., Chung, T.D.Y., et al., Eds.; Bethesda (MD), 2004). Being only a single cell line tested, the authors should have used an appropriate test to assess growth and proliferation. iv) The authors should provide an explanation for the peculiar result regarding the decreased cell proliferation with KO gene. v) There are no images showing morphological differences of cells.
6) The migration assay images have poor resolution. There is no scale bar. The cells are not visible. I suggest showing the images with and without the segmentation to better define the edges. The methodology of the wound healing assay is reported in references 35 and 36. Although the authors used supports μ-Dishes (Ibidi) in silicone inserts that allow an accurate assay execution for wound formation, I suggest adding some considerations about the limitations of the assay as wound coverage can be determined by increased proliferation rather than migration. In fact, additional assays are often suggested to evaluate migration and invasion more accurately (Pijuan J, et al. In vitro Cell Migration, Invasion, and Adhesion Assays: From Cell Imaging to Data Analysis. Front Cell Dev Biol. 2019 Jun 14;7:107. doi: 10.3389/fcell.2019.00107. PMID: 31259172; PMCID: PMC6587234).
7) The take home message of the work is unclear. I suggest explaining well and clearly how the bioinformatics results and in vitro evidence describe the possible prognostic role of the PCDHGC3 gene in GBM.
Author Response
Dear Reviewer,
Thank you for your valuable comments, which helped us to improve our manuscript. We have attempted to make all changes as indicated. Here is our point-by-point response:
Reviewer 1
A brief summary:
The authors of this manuscript describe the prognostic role of the PCDHGC3 gene in glioblastoma (GBM). First, they performed a bioinformatics analysis by The Cancer Genome ATLAS (TCGA) data, showing that the expression of PCDHGC3 is elevated in glioma and melanoma. Survival curves showed that patients with low PCDHGC3 expression have short progression free survival. According to these results, the authors evaluated PCDHGC3 expression on different human GBM cell lines, demonstrating that the U343 cell line had the highest expression. This line was chosen to generate the PCDHC3 knockout (KO) and to evaluate cell proliferation and cell migration.
Broad comments:
Experimental evidence with PCDHGC3 KO cell line have serious flaws that cannot be included in the publication in the current version.
Specific comments:
11) In the abstract, it would be appropriate to specify the term PCDHGC3 as a member of one of the clusters.
Response: thank you for this comment, we have specified it: “Here we examined the mRNA expression of PCDHGC3, a member of the PCDH g cluster, in non-cancerous brain tissue and in gliomas of different World Health Organization (WHO) grades and correlated it with the clinical data of the patients.”
2) Lines 81-82: authors declare “[..] to determine whether the PCDHGC3 expression may serve as a predictive biomarker”. A predictive biomarker gives information about the effect of a therapeutic intervention, but the authors did not consider any therapeutic association, therefore I suggest considering the aim of study at analyzing the PCDHGC3 as prognostic biomarker.
Response: We apologize for our mistake and have implemented the changes as kindly suggested by the reviewer.
3) Line 189: the authors state “Our results indicate that high PCDHGC3 mRNA expression levels correlate with longer PSF and OS in patients”. Survival curves analysis did not demonstrate significance for OS. Therefore, I suggest changing this sentence.
Response: We thank the reviewer for pointing our attention to this inaccuracy an have changed the sentence to “Our results indicate that high PCDHGC3 mRNA expression levels are associated with longer PSF and might be an independent predictor of OS as suggested by multivariable cox proportional hazard model.” (line 208)
4) The authors could perform a densitometric analysis of western blot to confirm the expression of the gene in GBM cell lines.
Response: Thank you for this suggestion. We have performed a densitometric analysis and show it now next to the Western blot image in Figure 5 (formerly Figure 4).
5) The authors show the MTT results plotting the number of cells in relation to the PCDHGC3 gene WT/KO. This result has serious errors because i) the absorbance data are not showed; ii) the MTT does not provide the number of cells, therefore the data are incorrect; iii) although this test is widespread for survival evaluation, it is not the ideal assay to evaluate the proliferative effect; indeed MTT is not reflecting necessarily cell proliferation and growth, but viable cell metabolism (Riss, T.L.; Moravec, R.A.; Niles, A.L.; Duellman, S.; Benink, H.A.; Worzella, T.J.; Minor, L. Cell Viability Assays. In Assay Guidance Manual, Markossian, S., Sittampalam, G.S., Grossman, A., Brimacombe, K., Arkin, M., Auld, D., Austin, C.P., Baell, J., Caaveiro, J.M.M., Chung, T.D.Y., et al., Eds.; Bethesda (MD), 2004). Being only a single cell line tested, the authors should have used an appropriate test to assess growth and proliferation. iv) The authors should provide an explanation for the peculiar result regarding the decreased cell proliferation with KO gene. v) There are no images showing morphological differences of cells.
Response: we thank the reviewer for this valuable comment. We agree that the MTT assay only shows the metabolic activity of the cells in accordance with cell number. Due to different absorbance values between independent experiments, we show the normalized values. As the reviewer suggested, we use now “metabolic activity” instead of “growth rate” and “proliferation”. We present new data in Figure 6 where we analyzed gene expression of genes involved in Wnt signaling. This result partially explains the lower metabolic activity observed in the MTT assay. The cells showed no morphological differences, therefore we decided not to show this images.
6) The migration assay images have poor resolution. There is no scale bar. The cells are not visible. I suggest showing the images with and without the segmentation to better define the edges. The methodology of the wound healing assay is reported in references 35 and 36. Although the authors used supports μ-Dishes (Ibidi) in silicone inserts that allow an accurate assay execution for wound formation, I suggest adding some considerations about the limitations of the assay as wound coverage can be determined by increased proliferation rather than migration. In fact, additional assays are often suggested to evaluate migration and invasion more accurately (Pijuan J, et al. In vitro Cell Migration, Invasion, and Adhesion Assays: From Cell Imaging to Data Analysis. Front Cell Dev Biol. 2019 Jun 14;7:107. doi: 10.3389/fcell.2019.00107. PMID: 31259172; PMCID: PMC6587234).
Response: thank you for this comment. We increased the size of the images in the migration assay. For space resons we show the segmentation and we think now it is now clearly visible that the space between the cell edges is empty. We indicate the magnification used in the legend and point out the need for further experiments:
“Further experiments are needed to elucidate the proliferation and invasiveness of KO cells.”
7) The take home message of the work is unclear. I suggest explaining well and clearly how the bioinformatics results and in vitro evidence describe the possible prognostic role of the PCDHGC3 gene in GBM.
Response: we have shortened the “take home message” and hope it is clearer now:
“For the first time, we analyzed PCDHGC3 expression in gliomas under consideration of their WHO grading. The overexpression of PCDHGC3 in gliomas grade 2/3 and GBM compared to non-cancerous brain samples was analyzed on both, mRNA and protein level. The correlation with clinical parameters indicated that PCDHGC3 might serve as a useful future marker for PFS in GBM patients. PCDHGC3 KO leads to changes in phenotype and gene expression in the U343 cell line, indicating its distinct role in GBM pathology.”

Reviewer 2 Report
The manuscript by Feldheim and Colleagues reports a potentially interesting result, that is the overexpression of an otherwise tumor-suppressor gene, Protocadherin Gamma C3 (PCDHGC3) in glioma of different grade.
However, in some instances the manuscript is not clear and must be improved, as it should be the design of the study, the methods used and the statistical analysis.
1. In the Results section, from line 125 onwards, please add the number of patients and the type of statistical test used, and justify discrepancies: at line 145, 35-40 GBM patients were used to generated Kaplan-Mayer plots, while at line 106/115 they are 46, and at line 139 they are 60. Please justify. I understand that from line 106 onwards, the Authors are reporting data from their own patients. Please make the reader aware of the work you have done.
2. On methods: why the study was done on mRNA and not on protein? Since proteins are much more stable than mRNA. This is particularly relevant because control tissue was obtained from autopsies: mRNA degrades within minutes from death, but for autopsies the time between death and sampling lasts hours, at best. Hence, it is not surprising to find an upregulation compared to controls. Did the Author have a positive control, that is higher in ‘normal’ compared to tumor samples, to verify the robustness of their method?
3. Concerning statistics: in different points the statistics is not clear. Line 142: Kruskal-Wallis test is the non-parametric homologue of ANOVA for multiple groups: hence it says that there is a difference AMONG ALL the groups, but it cannot say anything about the difference between couple of values (that are indicated above the bars of the histograms): the p<0.00x above histograms does not come from KW test, but from post-hoc test conducted on significant KW analysis: which post-hoc test was done to calculate the individual p? Bonferroni correction is not a post-hoc. Figure 3: assigning patients to two groups based on a single value (the median) is quite risky: to which class were assigned the sampled that had a median value? Is it biologically relevant to assign to ‘high’ or ‘low’ classes values around the median, that may differ by a single number? Could the lack of significance be possibly due to the addition of similar cases around the median in the two groups? Lines 179-180: “a significant correlation between …and…was found when the WMW test was used”: to my knowledge, the Wilcoxon-Mann-Whitney test is not a correlation test. It allows for the possibility that cases in one group are more correlated between themselves than those in the control group: hence it does give information on correlation between the variables ‘mRNA expression’ and ‘OS’.
4. Concerning results: PCDHGC3 has been described as a tumor suppressor and downregulates Wnt- and mTOR pathway (line 71): how the present data relate to this information? Is there any change in Wnt- pathway in the KO cells? Where are the data on expression? Only a single band is shown in fig. 4A, but it should be done in triplicate and correctly quantified relative to housekeeping gene expression: U 343 beta-actin appears much more intense than for the other cell lines. It is not clear whether technical or biological replicates were done. Were cells newly knocked down for different experiments and then used for western blot? Or cells were knocked down once and then plated in different dishes?
5. Discussion: please omit discussion on non-significant data. The number of cases is low in the different groups, hence I am worried about the true meaning of data. Can the Authors estimate the sample size based on the magnitude of the effect?
6. It would be valuable to add data on Wnt signaling downregulation (lines 228-230), which may explain the puzzling results obtained with KO cells. Otherwise, they remain too preliminary to be published.
As it stands, the manuscript describes a potentially interesting finding, yet the concerns I raised should be addressed by the Authors by adding more data.
MINOR:
Chapter 2: Results: the number of sub-chapters is wrong: 2.1, 2.2 etc., not as it is now: 1.1, 1.2 etc.
Line 106: to increase the readability, add some info on the ’60 patients’ (collected where? When?), otherwise the reader has to jump to the ‘Methods’ section to understand, since it is not clear that these are collected from the Authors and not a subset of the TCGA atlas.
Line 252: patients’ survival?
Author Response
Dear Reviewer,
Thank you for your valuable comments, which helped us to improve our manuscript. We have attempted to make all changes as indicated. Here is our point-by-point response:
Reviewer 2
The manuscript by Feldheim and Colleagues reports a potentially interesting result, that is the overexpression of an otherwise tumor-suppressor gene, Protocadherin Gamma C3 (PCDHGC3) in glioma of different grade.
However, in some instances the manuscript is not clear and must be improved, as it should be the design of the study, the methods used and the statistical analysis.
- In the Results section, from line 125 onwards, please add the number of patients and the type of statistical test used, and justify discrepancies: at line 145, 35-40 GBM patients were used to generated Kaplan-Mayer plots, while at line 106/115 they are 46, and at line 139 they are 60. Please justify. I understand that from line 106 onwards, the Authors are reporting data from their own patients. Please make the reader aware of the work you have done.
Response: We thank the reviewer for pointing our attention to the insufficient explanation behind the number of patients. In total, we determined the PCDHGC3 mRNA expression in 60 GBM patients. However, we were only able to retrospectively collect data from 46 of these patients, therefore we did only include these 46 patients for all following analyses. The only exceptions are the survival analyses, where we had to exclude further patients: In six patients’ clinical course we identified significant external confounders on survival (e.g. the decision to favor “experimental” therapeutic concepts or severe complications unrelated to the GBM) and therefore excluded those patients. Out of the remaining 40, five patients (mainly with a multifocal tumor growth) died or changed their therapeutic concept to “best supportive care” before matching the radiological RANO criteria for a progress/recurrence.
Furthermore, we clarified that the data presented from line 106 onwards is based on our own patient cohort by rephrasing the first sentence of the paragraph:
“The clinical data as well as the course of the disease and therapy could be collected for 46 of the 60 GBM patients of our own collective that were tested for their PCDHGC3 mRNA expression” (lines 108-110.)
- On methods: why the study was done on mRNA and not on protein? Since proteins are much more stable than mRNA. This is particularly relevant because control tissue was obtained from autopsies: mRNA degrades within minutes from death, but for autopsies the time between death and sampling lasts hours, at best. Hence, it is not surprising to find an upregulation compared to controls. Did the Author have a positive control, that is higher in ‘normal’ compared to tumor samples, to verify the robustness of their method?
Response: We absolutely agree with the reviewer that this is indeed a topic of high relevance and thank him/her for the opportunity to clarify the reasons for our decision. Firstly, we started our investigations by bioinformatical analyses. Currently, the data foundation of publicly available databases on trancriptome data on GBM is, to our mind, superior and more extensive compared to proteome data (e.g. there is to the best of our knowledge no comparable dataset to the IVY-GAP database). Therefore, our initial observations were based on mRNA and we also continued transcriptome analyses on our own collective for reasons of consistency. Secondly, with the methods available to us, we needed less tissue to accurately determine the mRNA compared to the protein expression, which was necessary as there was only little amount of specimen available in some cases (e.g. after stereotactic biopsy).
Also, we agree with the reviewer that unstable mRNA represent a major issue in the conduct of this project, however we believe that we did sufficiently address this. Tissue for mRNA extraction was frozen in liquid nitrogen directly after extraction and carefully treated afterwards. Also, we determined the mRNA expression by the ddCT-method using 18S mRNA as a reference for the first step of calculation and thereby making a bias by mRNA degradation nearly impossible. Lastly, we did not only use reference tissue obtained by control autopsies but also by biopsies of non-cancerous brain tissue (e.g. epilepsy surgery) to bypass that our data might be influenced by (un-)known confounders in either type of tissue. However, both the autopsy and biopsy tissue displayed a similar PCDHGC3 mRNA expression and were therefore combined to one group. We hope that this clarifies the reasons behind our project design in a satisfactory manner.
In addition, we now present in Figure 4 new data with protein analysis of PCDHGC3 in non-cancerous (n = 7) and GBM (n = 12) samples, confirming our results at the mRNA level.
- Concerning statistics: in different points the statistics is not clear. Line 142: Kruskal-Wallis test is the non-parametric homologue of ANOVA for multiple groups: hence it says that there is a difference AMONG ALL the groups, but it cannot say anything about the difference between couple of values (that are indicated above the bars of the histograms): the p<0.00x above histograms does not come from KW test, but from post-hoc test conducted on significant KW analysis: which post-hoc test was done to calculate the individual p? Bonferroni correction is not a post-hoc. Figure 3: assigning patients to two groups based on a single value (the median) is quite risky: to which class were assigned the sampled that had a median value? Is it biologically relevant to assign to ‘high’ or ‘low’ classes values around the median, that may differ by a single number? Could the lack of significance be possibly due to the addition of similar cases around the median in the two groups? Lines 179-180: “a significant correlation between …and…was found when the WMW test was used”: to my knowledge, the Wilcoxon-Mann-Whitney test is not a correlation test. It allows for the possibility that cases in one group are more correlated between themselves than those in the control group: hence it does give information on correlation between the variables ‘mRNA expression’ and ‘OS’.
Response: We apologize for the inaccurate information and wording. As the reviewer correctly presumed we used the Kruskal-Wallis test as an omnibus test and the Dunn test with Bonferroni correction as post-hoc test. We revised the manuscript adding the name of the Dunn test in lines 104, 143, 148, 401,.
Also, we agree with our reviewer that assigning patients to two groups based on the median poses some risk. However, to make general observations about possible association between PCDHGC3 and patient’s survival by Kaplan-Meier it was necessary to divide our collective. While Receiver operating characteristic (ROC) might have been an alternative to determine an ideal threshold, it also poses a certain risk of misinterpreting an effect that is caused by a statistical coincidence. Similarly, dividing the collective in more than two groups does not appear sensible due to the sample number. Therefore, we chose the median value to assign groups as it provides a first insight into possible effects with a low risk of overinterpreting results. We included an even number (n=40) of patients into OS analysis and there was a difference in mRNA expression between patients 20/21 locating the median inbetween and therefore allowing a clear distribution.
We also thank the reviewer for calling our attention to our inaccurate wording concerning the Wilcocon-Mann-Whitney test. We accordingly changed the sentence to “a significant association between … and .. was found when the WMW test was used”. (l.189)
- Concerning results: PCDHGC3has been described as a tumor suppressor and downregulates Wnt- and mTOR pathway (line 71): how the present data relate to this information? Is there any change in Wnt- pathway in the KO cells? Where are the data on expression? Only a single band is shown in fig. 4A, but it should be done in triplicate and correctly quantified relative to housekeeping gene expression: U 343 beta-actin appears much more intense than for the other cell lines. It is not clear whether technical or biological replicates were done. Were cells newly knocked down for different experiments and then used for western blot? Or cells were knocked down once and then plated in different dishes?
Response: We have now quantified these Western blot results showing the corresponding graph (Figure 5a). in addition, we present new data in Figure 6, where we analyzed gene expression of genes involved in Wnt signaling. We used different passages of a KO clone in the experiments.
- Discussion: please omit discussion on non-significant data. The number of cases is low in the different groups, hence I am worried about the true meaning of data. Can the Authors estimate the sample size based on the magnitude of the effect?
Response: We are aware of the limitations of our data but still think this is a valuable first report considering the role of PCDHGC3 in GBM.
- It would be valuable to add data on Wnt signaling downregulation (lines 228-230), which may explain the puzzling results obtained with KO cells. Otherwise, they remain too preliminary to be published.
Response: We present new data in Figure 6, where we analyzed gene expression of genes involved in Wnt signaling.
As it stands, the manuscript describes a potentially interesting finding, yet the concerns I raised should be addressed by the Authors by adding more data.
MINOR:
Chapter 2: Results: the number of sub-chapters is wrong: 2.1, 2.2 etc., not as it is now: 1.1, 1.2 etc.
Line 106: to increase the readability, add some info on the ’60 patients’ (collected where? When?), otherwise the reader has to jump to the ‘Methods’ section to understand, since it is not clear that these are collected from the Authors and not a subset of the TCGA atlas.
Line 252: patients’ survival?
Response: thank you for your valuable comments. We changed it in the manuscript.
Round 2
Reviewer 1 Report
Critical points of the work have been solved.
Reviewer 2 Report
The Authors addressed my concerns